# Mind the Gap: Diagnosing Spatial Reasoning Failures in Vision-Language Models

## Abstract

Vision-Language Models (VLMs) have captivated the research community by effectively merging visual and textual information, implying a holistic comprehension of the environment. These models find applications in tasks such as Image Captioning and Visual Question Answering, fostering the assumption that they perceive reality in a way similar to human cognition. However, this apparent understanding may be misleading. We argue that a critical component of comprehension—spatial reasoning—has been insufficiently addressed, as current benchmarks often conflate visual recognition with spatial reasoning, or focus on static properties rather than the dynamic simulation required for genuine spatial logic. In this study, we aim to address this limitation through a targeted diagnostic approach. Drawing from the fundamental elements of human cognition, we developed a curated evaluation suite designed to isolate the essential components of spatial reasoning: relational understanding, orientation, mental rotation, and visualization. We evaluated 17 state-of-the-art VLMs across a strictly controlled set of 1800 samples, split between synthetic settings and real-world images. Results indicate a substantial gap in performance: the apparent competence of these models decreases significantly under spatial reasoning tasks that require any dynamic transformation and manipulation of spatial information. On average, their performance parallels random guessing, which highlights a major systematic weakness in spatial reasoning in current VLMs. In addition to providing evidence for this limitation, this study provides the research community with a foundational diagnostic framework for probing model capabilities regarding spatial properties in their environment.

## 1 Introduction

Vision-Language Models (VLMs) have demonstrated impressive proficiency across a broad spectrum of multimodal tasks, such as Image Captioning, Visual Question Answering, and text-image retrieval (Liu et al., 2023; Dubey et al., 2024; Radford et al., 2021). Leveraging extensive datasets, these models effectively map intricate interactions between visual and textual data. However, one crucial facet of intelligence remains notably deficient: **spatial reasoning**. This essential skill entails understanding object locations, orientations, and their interrelations within a scene—a capability that is instinctive to humans but poses a substantial challenge for modern deep learning models (Zhang et al., 2025; Shiri et al., 2024; Chen et al., 2024a; Cheng et al., 2024).

Spatial reasoning is not a niche skill; it is fundamental to cognition. Developing in humans between the ages of two and eleven Hodgkiss et al. (2021), it underpins our ability to navigate and interact with complex environments (Johnson, 1987; Newcombe & Huttenlocher, 2000). Bridging this gap in AI is crucial for moving beyond static pattern recognition toward a human-like understanding of the physical world, a prerequisite for applications in robotics and autonomous navigation where agents must adapt to dynamic spaces (Venkatesh et al., 2021).

Despite the breadth of existing benchmarks, a critical diagnostic gap remains: the distinction between *static spatial perception* and *dynamic spatial simulation*. Although frameworks like SAT Ray et al. (2024) and MindCube Yin et al. evaluate spatial reasoning, they often conflate a model's ability to describe a fixed scene with its ability to mentally manipulate it. Similarly, VSI-Bench Yang et al. (2025) identifies a "reasoning bottleneck" in video, but its focus on complex indoor scenes makes

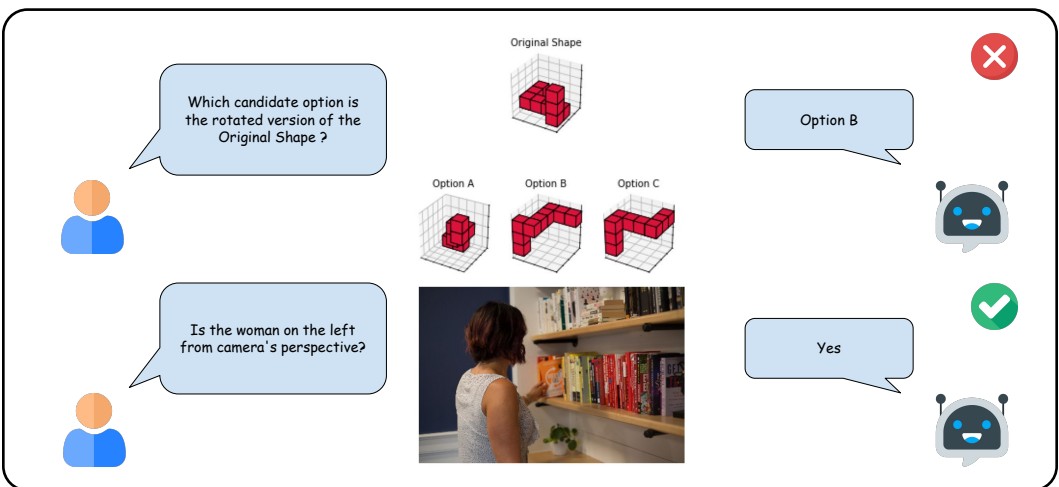

Figure 1: **The Static-Dynamic Dissociation.** Models frequently struggle to identify legitimate rotations, exposing deficiencies in tasks requiring dynamic internal simulation. However, they exhibit competence in discerning spatial relations present within static images.

it difficult to isolate whether failures stem from visual noise or a lack of cognitive machinery. Current evaluations fail to explain *why* models struggle: Is it a failure to parse the scene (Perception), or a failure to run the internal physics engine required to predict change (Simulation)? Without separating these modalities, the community risks overestimating VSI capabilities based on high performance in static recognition tasks.

To address this, we introduce **SRBench**, a cognitive psychology-based diagnostic suite designed to disentangle these capabilities. Unlike large-scale generalist benchmarks, our framework prioritises the isolation of reasoning variables. We adapt the gold-standard human cognitive tests—specifically the Mental Rotation Test (MRT) and the Paper Folding Test—to systematically assess models across four distinct pillars: (1) mental rotation, (2) spatial visualisation, (3) relational understanding and (4) egocentric navigation.

Table 1: Comparison of Related Spatial Reasoning Benchmarks. Unlike prior works which focus on video or synthetic environments, SRBench adapts psychometric standards to systematically disentangle static perception from dynamic simulation.

| Benchmark | Domain | Key Focus | Key Limitation/Insight |
|---|---|---|---|
| MindCube Yin et al. | Multiview | Mental simulation & "Map-then-reason" | Scaffolding maps improves reasoning. |
| SAT Ray et al. (2024) | Synthetic (ProcTHOR) | Procedural data (175k pairs) | Evaluation limited to LLaVA variants. |
| VSI-Bench Yang et al. (2025) | Video (Indoor) | Visual-spatial intelligence bottleneck | Models form fragmented local models. |
| OmniSpatial Jia et al. (2025) | Video & Image | Cognitive taxonomy (Psychology) | Covers outdoor dynamic scenes. |
| STARE Li et al. (2025b) | 3D Tasks | Multi-step visual simulation | Near-random performance on complex 3D tasks. |
| 11Plus Li et al. (2025a) | Aptitude Tests | Human vs. Model cognitive profiles | Compares human response time to model effort. |
| **SRBench (Ours)** | **Psychometric (Image)** | **Disentangling Perception vs. Simulation** | **Static-Dynamic dissociation across 17 models.** |

Our evaluation of 17 state-of-the-art VLMs reveals a fundamental fracture in current capabilities. We observe a **Static-Dynamic Dissociation**: while models exhibit strong performance on static tasks (e.g., Orientation, Spatial Relations), they suffer a catastrophic collapse on tasks requiring dynamic simulation. As illustrated in Figure 1, on mental rotation tasks, nearly all models—including GPT-4o—perform at or below random chance. This indicates that current models operate as surface-level observers rather than world simulators; they can describe what is, but cannot imagine how it changes.

In summary, our contributions are as follows:

- **Psychometric Grounding for VLMs:** We introduce SRBench, a curated suite of 1,800 examples stratified across four cognitive pillars. By adapting established psychometric paradigms (MRT, Paper Folding), we provide a rigorous testbed that isolates specific cognitive primitives rather than confounding them with visual noise.

- **The "Static-Dynamic" Dissociation:** We empirically demonstrate that modern VLMs possess a sharp divide between static perception and dynamic reasoning. While capable

of parsing spatial relations in fixed images, they fail to run the internal simulations required for mental manipulation.

- **Limits of Scaling on Simulation:** Through an extensive evaluation of 17 models (including GPT-4o, o1, and the InternVL-3/Qwen2.5 families), we show that scaling parameters improves articulated reasoning and static perception but yields diminishing returns on dynamic simulation. We argue that without specific architectural inductive biases for 3D continuity, even the largest models struggle to "imagine" object transformations reliably.

## 2 CONSTRUCTING THE BENCHMARK AND EXPERIMENTAL SETUP

Human spatial reasoning emerges from the intricate interplay of several cognitive abilities that allow us to navigate, manipulate, and understand our three-dimensional world (Hegarty, 2010; Darken et al., 1999; Wang & Spelke, 2002). Unlike previous benchmarks that evaluate isolated aspects of spatial cognition Ma et al. (2024); Kamath et al. (2023), our comprehensive evaluation framework systematically assesses Vision-Language Models across the fundamental interconnected pillars of human spatial reasoning: mental rotation, spatial visualization, relational understanding, and egocentric navigation.

### 2.1 MENTAL ROTATION

We begin by evaluating a model's capability for mental rotation—the ability to mentally transform three-dimensional objects in space. Drawing from the seminal Mental Rotation Test (MRT) (Cooper, 1975), which has served as the gold standard for measuring this cognitive ability in humans for decades, we adapt this classical paradigm to modern VLMs.

The original MRT presents participants with pairs of 3D objects or letters, rotated along various axes, challenging them to distinguish between identical shapes and their mirror images (Shepard & Metzler, 1971). Human performance is typically assessed through both accuracy and response time at rotation angles of 0°, 60°, 120°, and 180° (F Caissie et al., 2009).

Our digital adaptation follows this established protocol while accommodating the unique characteristics of VLMs. We manually craft five distinct polycube shapes and construct test images that feature the target shape in the top row, accompanied by four candidate shapes below. Among these candidates, exactly one represents the original shape rotated by 0°, 60°, 90°, or 120°; the remaining three consist of two mirrored versions at different rotations and one randomly selected unrelated shape.

To systematically vary the difficulty of the task, we develop two complementary variants. The *MRT-Hard* subset presents white shapes against blank backgrounds, offering minimal visual cues and posing a significant challenge to the model's internal spatial representations. Recognising that this austere presentation might limit model performance, we create the *MRT-Easy* subset, which incorporates coloured shapes positioned within a 3D Cartesian grid background and reduces the choice set to three candidates by removing one mirrored candidate. Each subset consists of 200 carefully designed test cases, as illustrated in Figure 2 (a and b).

### 2.2 SPATIAL VISUALIZATION

Beyond object rotation, spatial reasoning demands the ability to mentally simulate complex geometric transformations. We assess this through an adaptation of the Paper Folding Test (Ekstrom & Harman, 1976; McGee, 1979)—a psychometric instrument whose performance is strongly correlated with success in spatially demanding fields such as engineering and architecture (Carroll, 1993).

Each instance presents a temporal sequence of transformations: a paper square undergoes one or two folds (vertical, horizontal, or diagonal), followed by punching one to three holes through the folded configuration. The model must then predict the resulting hole pattern when the paper is unfolded, selecting from three plausible alternatives. This task directly probes the model's capacity to internalise sequential geometric operations and mentally simulate their cumulative effects—a

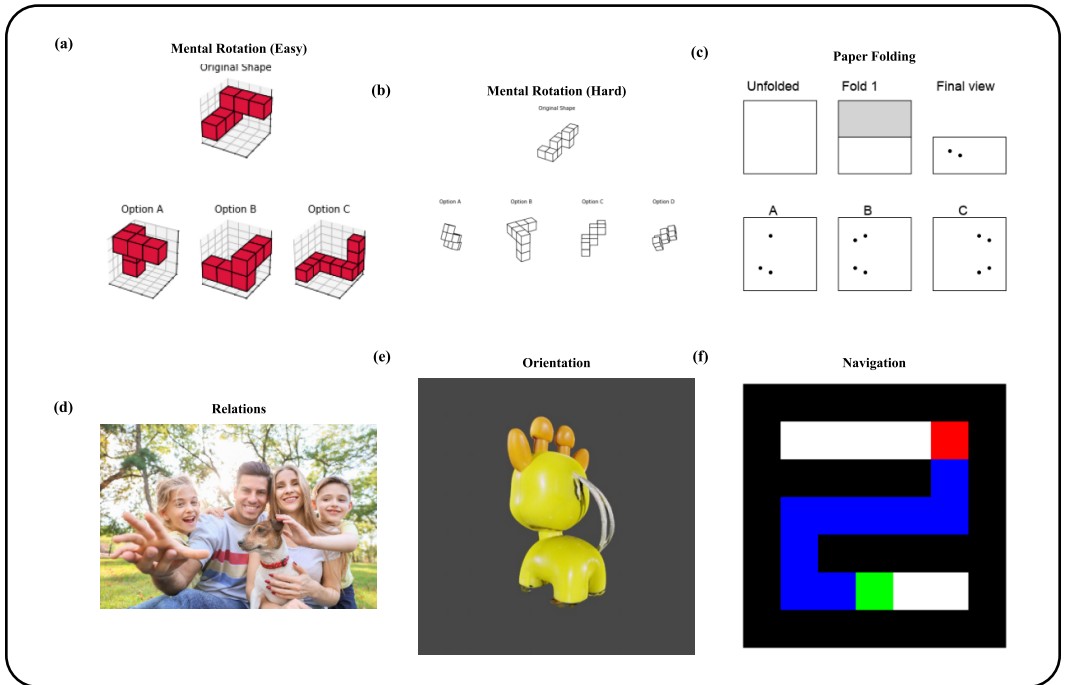

Figure 2: Representative image examples from SRBench spatial reasoning tasks. (a-b) Mental rotation tasks with hard and easy difficulty levels. (c) Paper folding visualization task. (d) Spatial navigation with route planning. (e) Spatial orientation and perspective-taking. (f) Spatial relations between geometric elements. Each panel demonstrates the visual complexity and cognitive demands of the respective spatial reasoning category in the benchmark dataset. More details can be found in Appendix B

cornerstone of spatial visualisation ability. The subset comprised 200 test cases, illustrated in Fig. 2 (c) as examples.

## 2.3 SPATIAL RELATIONS

Understanding the relative positioning and interactions between objects is the foundation of scene comprehension. We evaluate this critical capability using a curated sample from the Spatial-Obj dataset (Shiri et al., 2024), a rigorously constructed benchmark that contains 2,000 multiple choice queries regarding spatial relationships in natural images.

The authors generated this dataset employing an in-depth dual-stage annotation procedure, thoroughly encompassing 36 essential spatial relationships. These range from elementary positional notions such as 'right of' and 'above', to intricate geometric interactions including 'attached to,' 'touch', and 'overlapping'. The queries encompass diverse visual challenges including the identification of the precise location of the object, discrimination in orientation, and contextual spatial reasoning, providing a robust assessment of how well VLMs comprehend relational spatial language in realistic visual scenarios. This subset contains 400 test cases, with examples shown in Fig. 2 (d).

## 2.4 ORIENTATION AND NAVIGATION

Finally, we examine spatial reasoning within the critical domains of navigation and egocentric perspective taking, abilities essential for real-world spatial intelligence.

For navigation assessment, we employ the Maze-Nav component of SpatialEval (Wang et al., 2024), which challenges models to reason about paths through visual mazes represented by colored block configurations. Tasks include identifying routes from start (S) to exit (E) points, counting directional changes, and describing spatial relationships between key locations. While trivial for human spatial cognition, these challenges reveal significant limitations in current VLMs' navigational reasoning capabilities.

Complementing navigation assessment, we evaluate orientation understanding using 400 binary questions from EgoOrientBench (Jung et al., 2024). This benchmark addresses critical inconsistencies in spatial orientation evaluation by establishing a unified, camera-centric perspective framework. Through an eight-class egocentric taxonomy (Left, Right, Front-Left, Back-Right, etc.), it provides consistent object orientation definitions relative to the observer's viewpoint. This egocentric approach not only enhances evaluation reliability but also aligns with the increasing need for VLMs to operate effectively in user-centered, real-world applications, such as robotics, where spatial understanding must be grounded in human perspective. Each of this subsets contain 400 test cases, with examples shown in Fig. 2 (e and f).

## 2.5 SETUP

Our experiments were conducted with PyTorch (Paszke et al., 2019) and Hugging Face Transformers (Wolf et al., 2020). We evaluated the spatial reasoning capabilities of 17 VLMs, which include open-source and commercial models. Specifically, from the commercial side, we included evaluations of OpenAI's GPT-4o and o1 (Achiam et al., 2023; Jaech et al., 2024). The open source model set consists of: QwenVL2.5 in sizes 3B, 7B, 32B, and 78B (Bai et al., 2025); Llava 1.5 7B (Liu et al., 2024); LlavaNext 7B (Li et al., 2024); Idefics3 8B (Laurençon et al., 2024) and SmolVLM2 at 500M and 2.2B (Marafioti et al., 2025). Additionally, MiniCPM-V-2.6 8B (Yao et al., 2024); InternVL-3 models at 8B, 38B, and 78B (Chen et al., 2024b) and Gemma3 at 12B, and 28B. All models are instruction tuned and the experiments were conducted using greedy decoding (Germann, 2003) and Chain-of-Thought (Wei et al., 2022) prompting. For OpenAI's models, we used the Azure OpenAI API service, while for the open-source models, inference was performed using $2 \times$ H200 140GB NVIDIA GPUs.

## 3 RESULTS

Our investigation into the spatial reasoning of contemporary VLMs reveals a compelling, two-part narrative. On one hand, models exhibit a promising, emergent ability to parse static visual scenes. On the other, this competence proves remarkably brittle, collapsing entirely when confronted with tasks that require dynamic mental manipulation. This core tension, explored below, points to a fundamental gap between superficial pattern recognition and robust spatial cognition.

### 3.1 EMPIRICAL COHERENCE OF THE BENCHMARK TASKS

To justify integrating these diverse spatial tasks under a single benchmark, we computed pairwise correlation coefficients across task performances (aggregated from model outputs and behavioural data) and performed hierarchical clustering (using Ward's linkage) to reveal shared latent factors. This empirical analysis builds on the theoretical foundations detailed in the task construction above, grounded in established cognitive paradigms.

As shown in Figure 3, tasks cluster into meaningful subgroups: MRT Easy and Hard (r=0.73) reflect rotation centred on objects; Orientation and Relations (r=0.87) capture egocentric perspectives; and Paper Folding and Navigation (r=0.74) involve sequential transformations. Moderate cross-cluster correlations (e.g. 0.59 between Orientation and Paper Folding) support their aggregation as components of a unified spatial cognition construct, while low/negative ones (e.g. -0.11 between MRT Hard and Orientation) highlight diagnostic distinctions. This empirical structure demonstrates that the tasks are not an arbitrary collection but capture overlapping cognitive processes, with two major branches: small-scale object manipulation (MRT-dominant) versus large-scale environmental processing (Orientation/Relations/Paper Folding/Navigation).

### 3.2 THE FRAGILITY OF SPATIAL INTELLIGENCE: FROM STATIC COMPETENCE TO DYNAMIC COLLAPSE

At first glance, the models detailed in Table 2 demonstrate a solid grasp of basic spatial properties. On static tasks like **Orientation** and **Relations**, leading architectures such as InternVL-3 38B achieve high accuracy (77.5% and 73.5%, respectively), suggesting they can adeptly identify and relate objects in a fixed scene This initial success, however, masks a profound underlying weakness. This apparent competence is undermined when models must perform internal simulations of

| Model | Paper Folding | MRT Easy | MRT Hard | Navigation | Orientation | Relations | Overall |
|---|---|---|---|---|---|---|---|
| *Open-Source Models* | | | | | | | |
| Random | 33.0 | 33.0 | 25.0 | 25.0 | 50.0 | 25.0 | 32.0 |
| Idefics3 8B | 35.0 | 28.0 | 22.5 | 30.5 | 64.0 | 59.8 | 43.35 |
| InternVL-3 8B | 27.0 | 33.0 | 28.5 | 18.3 | 69.5 | 66.3 | 43.64 |
| InternVL-3 38B | 42.5 | **40.5** | **29.0** | 43.0 | **77.5** | 73.5 | 55.00 |
| InternVL-3 78B | 43.5 | 34.5 | 23.0 | **55.0** | 74.2 | **73.8** | **55.77** |
| MiniCPM-V 2.6 | 35.5 | 32.5 | 24.0 | 23.5 | 47.0 | 41.0 | 34.65 |
| Qwen2.5-VL 3B | 24.0 | 29.0 | 21.1 | 19.3 | 60.3 | 53.8 | 37.50 |
| Qwen2.5-VL 7B | 36.0 | 35.0 | 26.0 | 21.0 | 65.0 | 63.2 | 49.25 |
| Qwen2.5-VL 32B | 42.5 | 34.0 | 22.5 | 42.25 | 68.5 | 69.25 | 50.94 |
| Qwen2.5-VL 72B | **45.0** | 39.5 | 24.0 | 40.8 | 69.5 | 73.3 | 52.33 |
| SmolVLM2 500M | 29.5 | 36.0 | 27.5 | 34.5 | 51.8 | 37.8 | 37.53 |
| SmolVLM2 2.2B | 35.0 | 31.9 | 11.0 | 17.8 | 65.2 | 43.4 | 36.38 |
| Gemma 3 12B | 31.0 | 32.0 | 24.0 | 22.3 | 57.5 | 29.95 | 34.10 |
| Gemma 3 27B | 16.50 | 22.50 | 16.00 | 25.00 | 57.2 | 47.3 | 34.3 |
| LLaVA-1.5 7B | 36.0 | 35.5 | 25.5 | 36.0 | 52.8 | 31.4 | 37.1 |
| LLaVA-NeXT 7B | 25.0 | 34.5 | 27.0 | 20.3 | 53.1 | 48.3 | 36.29 |
| *Proprietary Models* | | | | | | | |
| o1 (Undisclosed) | 36.0 | 33.0 | 20.5 | 33.3 | 71.0 | 64.8 | 47.05 |
| GPT-4o (Undisclosed) | 36.0 | 32.0 | 20.0 | 32.8 | 72.5 | 66.5 | 47.48 |

Table 2: Performance of models across various spatial reasoning tasks. Models are grouped into open-source and proprietary categories. All scores are accuracy percentages. The best performance in each category is highlighted in **bold**.

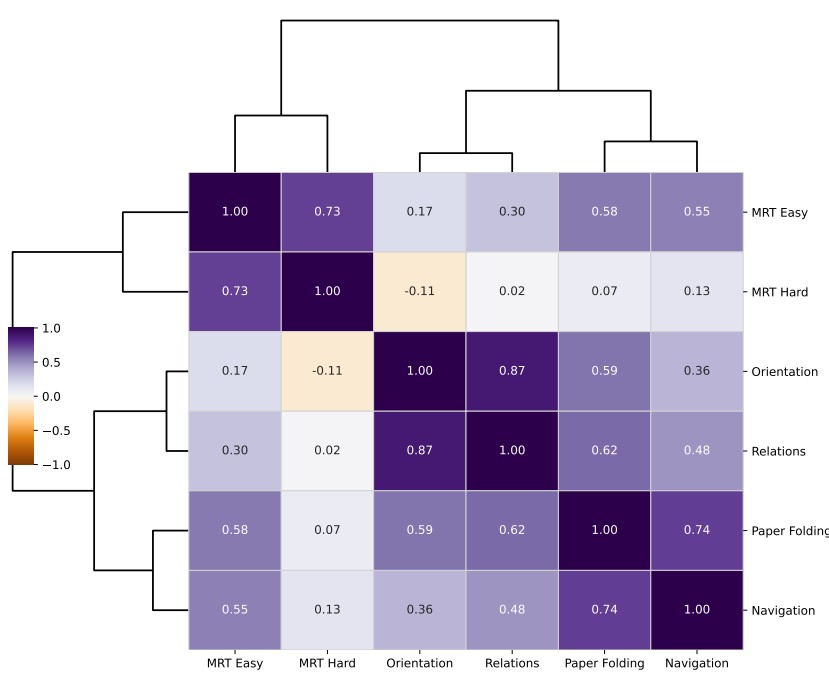

Figure 3: Hierarchical clustering of spatial tasks based on pairwise correlation coefficients, revealing subgroups of related cognitive processes (e.g., mental rotation vs. relational/scene-based reasoning). Correlations range from -1.0 (orange) to 1.0 (purple).

dynamic object transformations. On the **Mental Rotation Test (MRT Hard)**, a task requiring com-

plex, multi-axis mental manipulation, performance plummets. The failure is not merely a gradual decline, but a catastrophic collapse: most models do not outperform the random baseline. Most strikingly, even state-of-the-art models like GPT-4o score just $20\%$, performing significantly *worse* than random chance ($25\%$). Although mental rotation demands cognitive effort, the average adult accuracy in standard MRT tasks is typically well above 70% (Vandenberg & Kuse, 1978)—far exceeding the random baseline. This deficit likely stems from factors such as biases in training data (lack of rotated/diagonal views, spurious correlations), pretraining focused on static descriptions over internal simulations, and limitations in architecture for encoding continuous 3D priors. This indicates that their success in static spatial tasks does not imply the capacity to simulating transformations; they have learnt to describe the world as it is, but cannot reliably reason about how it might change Newman et al. (2024); Li et al. (2025c).

## 3.3 SCALING LAWS AND THE EMERGENCE OF ARTICULATED REASONING

As we scale models from billions to tens of billions of parameters, a distinct shift in the cognitive style emerges. Smaller models, such **as InternVL3-8B**, tend to produce concise and direct answers, offering little insight into their decision-making process. Their larger counterparts, such as **InternVL-78B**, behave fundamentally differently. They engage in articulated step-by-step reasoning, verbalising their analysis of visual evidence, and systematically evaluating options. This transition from opaque, "black-box" intuition to a more transparent, deliberative process suggests that scaling does not just improve accuracy—it unlocks more sophisticated and explicit reasoning pathways.

This qualitative evolution is mirrored by quantitative gains. Across the QwenVL2.5 and InternVL-3 families, models with tens of billions of parameters generally show much better performance compared to smaller ones (for example, **InternV-L3 78B** scores $55.77\%$ versus $43.64\%$ for the 8B variant). But scaling is not strictly monotonic: mid-size models sometimes beat larger ones (e.g., **InternVL-3 38B** outperforms 78B on the 'MRT hard' split), and we observe plateaus with little or no gain for some jumps, as depicted in Fig. 4. Given that model size typically covaries with various other elements, such as the training ensemble, objectives, data, and optimisation processes, it is not safe to assert that these effects arise solely due to the number of parameters. A plausible set of mechanisms that accompany scaling helps explain the qualitative shift. Larger parameter counts increase representational capacity, enabling models to internalize multi-step algorithms or templates for reasoning rather than relying on single-step heuristics. Larger models are also typically trained with more compute over longer runs on much bigger and more diverse corpora, raising the chance they encounter examples that demonstrate explicit, chain-of-thought–style analyses which they can imitate. These correlations necessitate controlled ablation studies to determine causality. Crucially, even the best and largest models still fail catastrophically on the hardest tasks, showing that scale alone does not resolve the underlying gaps in their reasoning toolkit.

## 3.4 A GRANULAR DISSECTION OF FAILURE MODES

To understand the limits of scaling, we performed a granular analysis of common failure modes. This investigation revealed a consistent Achilles' heel across all models and scales: a fundamental difficulty in processing diagonal and rotational transformations, particularly evident in the 'MRT hard' and 'Paper folding' tasks.

### 3.4.1 STATIC PERCEPTION VS. DYNAMIC & DIAGONAL REASONING

The most straightforward tasks reveal a foundational bias. In **Orientation** tasks, every model is more adept at identifying cardinal directions (e.g., "front") than diagonal directions (e.g., "front left"). This suggests an inbuilt preference for axis-aligned spatial judgments, a tendency that is highly likely attributable to biases in the training data.

This perceptual weakness extends to the **Relations** task, where reasoning accuracy declines. Through manual qualitative inspection of model responses, we observed that models reliably resolved queries involving static, unambiguous relationships (e.g., *"The bus is to the left of the building"*), yet their responses deteriorated noticeably for prompts involving agents performing actions (e.g., *"The man is **holding** the..."*). These observations suggest that while models can parse a static layout, they fail to build a robust model of interactions, a more complex and dynamic form of reasoning.

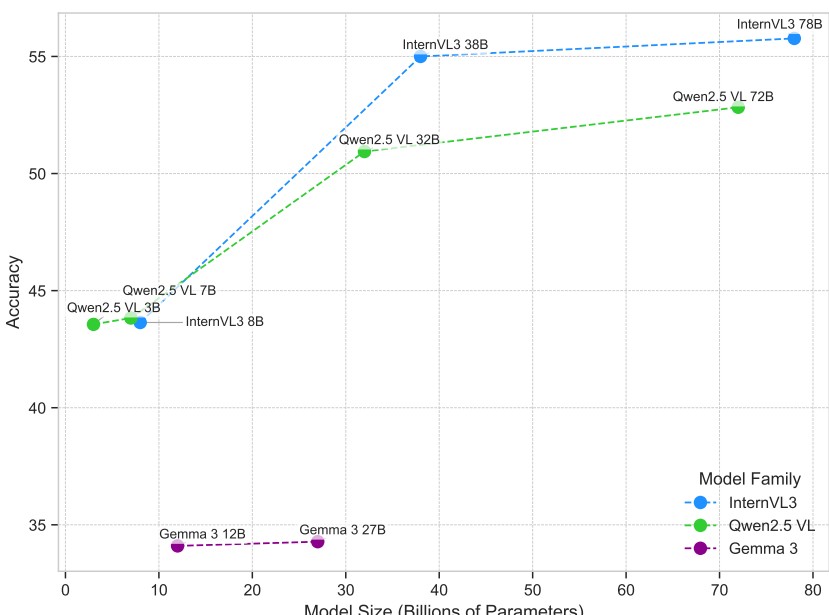

Figure 4: Model accuracy as a function of parameter count. While a positive trend exists within families, architectural differences create distinct performance tiers, highlighting that scale alone is not a panacea for complex reasoning failures

### 3.4.2 THE FRAGILITY OF ABSTRACT SPATIAL TRANSFORMATION

Qualitative analysis of model responses shows that difficulties with dynamic operations are most evident when models must mentally simulate transformations. In the **Paper Folding** subset, outputs were more reliable for simple axis-aligned folds than for diagonal ones, and reasoning quickly degraded as additional folds were introduced. When asked to track more than a few sequential folds, responses often became inconsistent or contradictory, suggesting that maintaining an object's state across transformations exceeds the models' effective reasoning capacity. A similar pattern appeared in the more demanding **MRT** tasks. For instance, the **InternVL-38B** model was most stable on medium-complexity objects, but its answers deteriorated as the number of polycubes increased. For medium-complexity shapes, the model generally counted polycubes correctly and reasoned more consistently; for more complex shapes, it often miscounted polycubes, leading to fabricated or inaccurate reasoning. This supports the view that model strategies handle only limited complexity and degrade predictably once that limit is crossed. These failures—from a bias against diagonals to difficulty tracking sequential rotations—indicate that achieving robust, human-like reasoning will require not just larger models, but new architectures and inductive biases tailored to dynamic object transformations. This conclusion is reinforced by performance on tasks such as navigation and 'MRT easy'. In the most difficult 'MRT hard' partition, Qwen2.5-VL models perform poorly, with only minor gains for the largest variants. Thus, while scaling generally improves analytical reasoning for moderately complex problems, it does not ensure better performance on tasks that exceed current architectural limits, where more explicit reasoning may offer little benefit and can even be detrimental.

## 4 RELATED WORK

Recent advances in Multimodal Large Language Models (MLLMs) have shifted focus from simple visual recognition to complex spatial reasoning. This section reviews concurrent benchmarks and frameworks that evaluate the ability of models to construct mental models, perform dynamic reasoning, and align with human cognitive processes.

### 4.1 3D Spatial Reasoning and Mental Simulation

Several works focus on the ability of models to internalise 3D spaces from limited observations. MindCube (Yin et al.) evaluates the ability of VLMs to construct spatial mental models through multiview observations. It employs questions that span three patterns of camera movement: rotation, around, and among to test the reasoning about occluded spaces. A critical finding from MindCube is the efficacy of the "map-then-reason" approach, where scaffolding VLMs to first generate explicit 2D cognitive maps prior to reasoning significantly outperforms passive map injection or view interpolation.

The concurrent work of STARE (Li et al., 2025b) highlights the fragility of current models in this domain. Their analysis reveals that while models perform well on simple 2D transformations, they struggle significantly with multi-step visual simulations in 3D tasks, often achieving near-random performance.

To address data scarcity in this domain, SAT Ray et al. (2024) introduces a procedural framework that uses the ProcTHOR simulator. SAT generates 175k synthetic QA pairs covering both static relationships and motion-based reasoning tasks (e.g., egocentric movement, object motion, and perspective shifts). However, it should be noted that the current evaluation of SAT is restricted to two LLaVA variants, leaving its impact on a wider range of architectures less explored.

### 4.2 Video-Based and Dynamic Spatial Intelligence

Moving beyond static imagery, recent benchmarks have begun to probe spatial intelligence in video. VSI-Bench Yang et al. (2025) represents a comprehensive effort in this space, evaluating MLLMs on more than 5,000 questions in 288 indoor videos. Their analysis identifies spatial reasoning—rather than visual perception or linguistic ability—as the primary bottleneck. Crucially, VSI-Bench finds that models tend to form fragmented local world models rather than unified global cognitive maps. However, VSI-Bench is limited to static indoor scenes with restricted camera motion.

Addressing these environmental limitations, OmniSpatial Jia et al. (2025) introduces a taxonomy grounded in cognitive psychology, categorising tasks into dynamic reasoning, complex spatial logic, spatial interaction, and perspective-taking. Unlike VSI-Bench, OmniSpatial evaluates models on both video and images and explicitly extends the domain to include dynamic outdoor environments.

### 4.3 Cognitive Alignment and Aptitude Testing

Finally, researchers are exploring how model reasoning processes align with human cognition. The 11Plus-Bench Li et al. (2025a) uses realistic 11+ aptitude tests to measure "cognitive profiles." By annotating instances with perceptual complexity and reasoning steps, this benchmark moves beyond coarse task-wise accuracy. Uniquely, it compares human response times directly against model token-level effort, allowing a granular analysis of the computational cost of reasoning relative to human cognitive load.

## 5 Conclusion

This paper studies spatial reasoning in VLMs—the ability to infer, predict, and manipulate geometric relationships and transformations (rotation, translation, scaling, occlusion) from images—by providing a clear definition, a robust benchmark with synthetic and real-world images, and an evaluation of 17 state-of-the-art VLMs. We find a stark gap: while most VLMs handle tasks that infer information present in an image, their performance falls to near-random on tasks that require reasoning about transformations, revealing a major limitation with important practical consequences. Our work takes a step toward addressing this gap; future research should analyze which cues models use in natural images, introduce inductive biases that explicitly encode transformations, and design architectures or modules for object-centric representation and manipulation of transformations. Continued study of how spatial components interact and how other visual cues support reasoning will be crucial to achieving more human-like spatial reasoning in AI models.

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

This appendix provides supplementary materials and detailed examples that support the main paper's findings. Section A offers a curated look at specific instances where leading models failed, revealing critical gaps in their spatial reasoning capabilities.

## A ILLUSTRATIVE EXAMPLES OF REASONING FAILURES

These case studies are not mere errors; they are windows into the systematic shortcomings of current models, providing tangible evidence for the key phenomena discussed in our paper. Each example includes the model, the task, the question, the model's (often incorrect) reasoning, the correct answer, and a brief analysis of the failure.

## A.1 THE PERCEPTUAL-ABSTRACTIVE GAP

This gap appears when a model can correctly identify individual objects and their properties but fails to synthesize them into a coherent, abstract understanding of the scene. It sees the trees but misses the forest.

### A.1.1 FAILURE ON DIAGONAL VS. CARDINAL ORIENTATIONS

Models show a distinct weakness in identifying diagonal orientations compared to simpler cardinal directions, pointing to a foundational bias for axis-aligned reasoning.

**Example 1: Diagonal Orientation Failure**

- **Model:** Gemma-3 27B-it
- **Task:** Orientation
- **Question:** Is the toy truck facing 'back left' from the camera's perspective? Answer with 'yes' or 'no' only.
- **Model's Prediction:** YES
- **Correct Answer:** NO

**Model's Reasoning:**

```
The camera is positioned to the side of the truck, and slightly
towards the front. The front of the truck (white cabin) is on
the left side of the image, and the back (green tank) is on
the right side of the image. Therefore, the truck is facing
towards the back and left from the camera's perspective.
```

**Analysis:** This is a classic failure of synthesis. The model correctly identifies the **perceptual ingredients** ('back' and 'left') but cannot combine them into the correct **abstract recipe** (the diagonal orientation 'back-left'). It treats the components as a simple checklist rather than an integrated spatial concept.

### A.1.2 FAILURE ON DYNAMIC ACTION-BASED RELATIONS

Models that succeed at identifying static relationships between objects often fail when the relationship is defined by a dynamic action performed by an agent.

**Example 2: Action-Based Relation Failure**

- **Model:** InternVL-3 8B
- **Task:** Relations
- **Question:** Is the person using the tool to hold the food on the left or the right side of the image?
- **Model's Prediction:** RIGHT
- **Correct Answer:** LEFT

**Model's Reasoning:**

```
The image shows a person holding a piece of food with a tool.
The food is located on the right side of the image.
```

**Analysis:** The model sees a static scene of objects—person, tool, food—but fails to comprehend the **action** ('using the tool to hold') that connects them. It correctly identifies the location of the food but completely misses the context of the question. It answers "where is the food?" instead

of "where is the *act of holding* happening?", highlighting a critical gap in reasoning about dynamic events.

## A.2 THE FRAGILITY OF ABSTRACT SPATIAL TRANSFORMATION

This category of failure occurs when models are tasked with mentally simulating changes to an object's state, such as folding or rotating it. Their ability to perform these "mental gymnastics" is surprisingly brittle.

### A.2.1 EXPONENTIAL FAILURE IN SEQUENTIAL TRANSFORMATION (PAPER FOLDING)

The most dramatic failures occur when models must track an object's state through multiple sequential transformations. Performance collapses exponentially with each additional fold.

**Example 3: 3-Fold Paper Folding Failure**

- **Model:** InternVL-3 78B (Top Performer)
- **Task:** Paper Folding
- **Question:** A piece of paper is folded three times, and one hole is punched. Which option shows the unfolded paper?
- **Model's Prediction:** D
- **Correct Answer:** B

**Model's Reasoning:**

```
After the first fold, there are 2 layers. After the second fold,
there are 4 layers. After the third fold, there are 8 layers.
A hole is punched through all 8 layers. When unfolded, there
will be 8 holes. Option D shows 8 holes.
```

**Analysis:** This reasoning is a textbook case of the Perceptual-Abstractive Gap applied to transformation. The model demonstrates a shallow, **quantitative success** (correctly calculating $2^3 = 8$ holes) while utterly failing the core **qualitative, spatial task** of predicting the holes' symmetrical pattern. It found an answer with the right *number* of holes but in the completely wrong *configuration*, revealing that its 'reasoning' was a brittle shortcut, not a true mental simulation of unfolding the paper.

### A.2.2 THE COGNITIVE "SWEET SPOT" IN MENTAL ROTATION

The top-performing model's accuracy on MRT tasks does not scale linearly with object complexity. It fails in overly simple and overly complex shapes, performing best in objects of medium complexity.

**Example 4a: Failure on a Simple Shape**

- **Model:** InternVL-3 78B
- **Task:** MRT Hard
- **Question:** (Image with a simple 5-cube shape)
- **Model's Prediction:** A
- **Correct Answer:** D

**Model's Reasoning:**

```
The original shape is composed of five red cubes. By rotating
the original shape 90 degrees to the right around the vertical
```

```
axis, it matches the configuration of Option A.
```

**Analysis:** The model's failure on a simple shape is revealing. It suggests that without sufficient complexity to anchor its analysis, the model resorts to **flimsy heuristics** that lead to error. It is akin to overthinking a simple problem and making a careless mistake because its more powerful, systematic reasoning pathways were not fully engaged.

**Example 4b: Success on a Medium-Complexity Shape**

- **Model:** InternVL-3 78B
- **Task:** MRT Hard
- **Question:** (Image with a more complex 7-cube shape)
- **Model's Prediction:** B
- **Correct Answer:** B

**Model's Reasoning:**

```
The original shape is an assembly of seven cubes. Let's trace
the key features. There is a central column of three cubes and
two arms protruding. By performing a 180-degree rotation around
the horizontal axis, the shape aligns perfectly with Option B.
```

**Analysis:** Here, with a moderately complex object, the model hits its stride. Its reasoning becomes more descriptive and structural ("central column," "two arms"), indicating a deeper level of engagement. It successfully simulates the rotation because the problem is complex enough to demand **genuine spatial processing** but not so complex as to overwhelm it. This success, contrasted with the failure on the simpler shape, perfectly illustrates the non-linear "sweet spot" for its performance.

## B   DETAILED EXAMPLES OF SRBENCH

In this appendix, we provide qualitative examples for the various tasks comprising SRBench. These visualisations illustrate the input modalities and the expected spatial reasoning required by the model. The benchmark covers three primary categories of spatial cognition: Object Manipulation, Spatial Relations, and Navigation/Orientation.

### B.1   SPATIAL VISUALIZATION AND MANIPULATION

Figures 5 and 6 illustrate the Mental Rotation Task. The model is presented with a reference object and a target object and must determine whether the target is a rotation of the reference or a distinct shape. Figure 5 demonstrates an "Easy" sample of difficulty, involving a single-axis rotation with minimal occlusion. Conversely, Figure 6 represents a "Hard" difficulty sample, requiring reasoning over multi-axis rotations and complex 3D structures.

**MRT Easy**

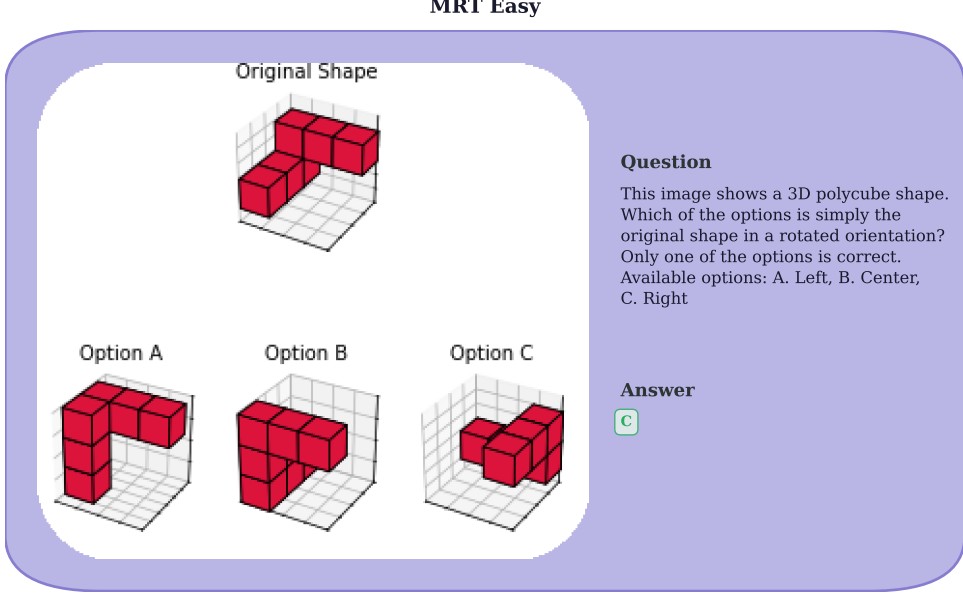

Figure 5: A sample from the **MRT (Easy)** subset. The target object requires a simple rotation along on[...]

**MRT Hard**

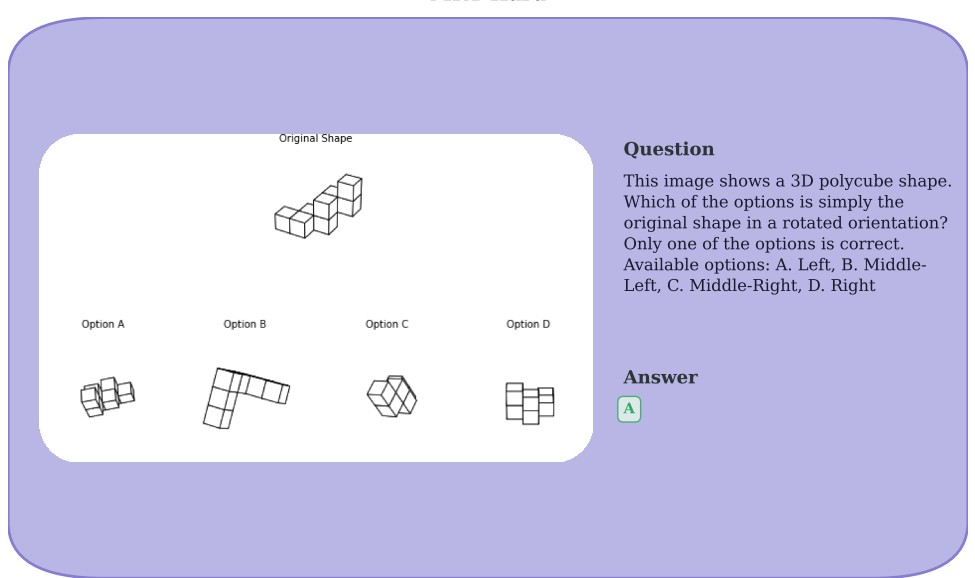

Figure 6: A sample from the **MRT (Hard)** subset. This task involves complex multi-axis rotations and higher structural complexity.

Figure 7 visualises the Folding task of paper. In this task, the model observes a sequence in which a 2D sheet is folded and potentially punched with holes. The model must mentally "unfold" the paper to predict the final 2D pattern or hole configuration, testing its capacity for non-rigid spatial transformations.

## B.2 SPATIAL RELATIONS

Figure 8 details the Spatial Relations task, evaluating the understanding of geometric predicates.

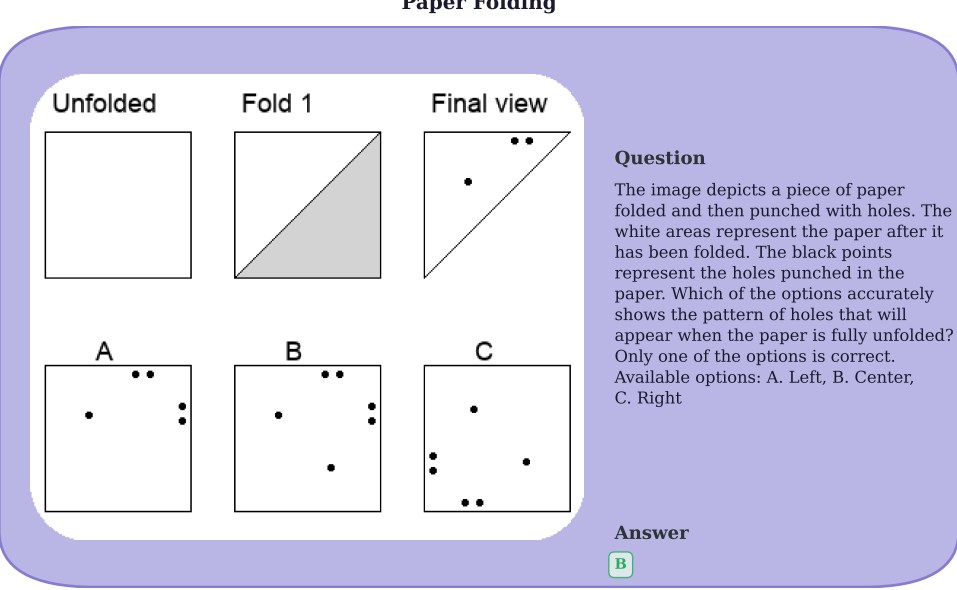

Figure 7: The **Paper Folding** task requires the model to mentally simulate the unfolding and subsequent perforation of a two-dimensional sheet and to infer the number of holes that will be present in the paper once it is fully unfolded.

**Relations**

Question

Is the woman on the left from camera's perspective? A. yes B. no

Answer

A. yes

Figure 8: Example of the **Spatial Relations** task. The model must identify the correct geometric predicate describing the relationship between the highlighted objects.

### B.3 NAVIGATION AND ORIENTATION

Figure 9 illustrates the Orientation task, which requires determining the direction of gaze of the subject depicted in the image. Figure 10 presents the Navigation task, which involves planning a trajectory through a maze-like environment.

**Orientation**

Figure 9: The **Orientation** task. The model is required to determine the directional orientation of the rhinoceros depicted in the image.

**Navigation**

Figure 10: The **Navigation** task. The figure illustrates a successful path execution from start (green) to goal (red) avoiding obstacles.

## C   USE OF LARGE LANGUAGE MODELS

In the preparation of this paper, Large Language Models (LLMs) were utilised to refine the text, improving clarity, grammatical precision, and stylistic flow without altering the substantive ideas or original authorship. This AI-assisted process allowed a more polished presentation of the research while maintaining academic integrity.

