# OpenReview forum: "Mind the Gap: Diagnosing Spatial Reasoning Failures in Vision-Language Models"
_ICLR.cc/2026/Conference — Submitted to ICLR 2026_

### Official Review · Reviewer_PJzY · 2025-10-29

**Soundness:** 1
**Presentation:** 2
**Contribution:** 1
**Rating:** 2
**Confidence:** 4

**Summary:**

This paper aims to look at the critical component of spatial reasoning based on inspirations from human cognition. It introduces a diagnostic framework designed to isolate components in spatial reasoning from relational, mental rotation to visualization. However, 17 state-of-the-art vision-language models (VLMs) show random chance results, suggesting that there's a major systematic weakness in spatial reasoning in current VLMs.

**Strengths:**

1. Paper is easy to follow.

**Weaknesses:**

1. The paper claimed that current benchmarks primarily test models’ ability to identify object positions rather than evaluate genuine spatial logic. However this is not true as many existing benchmarks has looked at spatial logic like object relations and mental rotation beyond positioning [1,2,3].
2. The paper claimed that it draws from the fundamental elements of human cognition. However some of the core tasks are challenging / unsolvable even for average humans. For example, the mental rotation tasks are very hard to solve for humans so it is not surprising that vision language models (VLMs) cannot solve it. Human evaluation with **non-biased participants** is necessary in the paper.

References

[1] Ray, Arijit, et al. "Sat: Spatial aptitude training for multimodal language models." arXiv e-prints (2024): arXiv-2412.

[2] Ma, Wufei, et al. "3dsrbench: A comprehensive 3d spatial reasoning benchmark." Proceedings of the IEEE/CVF International Conference on Computer Vision. 2025.

[3] Zhang, Weichen, et al. "Open3dvqa: A benchmark for comprehensive spatial reasoning with multimodal large language model in open space." arXiv preprint arXiv:2503.11094 (2025).

**Questions:**

1. Figure 2 should add detailed descriptions of each tasks like the prompting questions. It's hard to understand what the tasks are for Paper Folding and Orientation from looking at the figure.

---

> ### Author Response · Authors · 2025-11-20
>
> We thanks the reviewer for the feedback. Below we address the weaknesses mentioned.
>
> **Weakness:**
>
> 1.	We thank the reviewer for this suggestion. While other benchmarks also assess VLM spatial capabilities, a closer examination reveals that most reduce to position/orientation identification plus calculation rather than genuine spatial reasoning. For example, Open3D-VQA's failure analysis shows models fail primarily due to lacking 3D representations (depth estimates, pose detection) rather than logical reasoning deficits—questions like "Is A taller than B?" reduce to detecting heights and comparing numbers. Our benchmark targets the underlying spatial reasoning abilities that enable such tasks.
> 2.	Our tasks are intentionally challenging, as they are modeled after high-level cognitive and psychometric tests designed to probe the limits of reasoning. However, the claim that "some of the core tasks are challenging / unsolvable even for average humans." is false as the average performance on MRT tasks –which is the hardest– is around 70% [1]
>
> **Questions:**
>
> We will revise Figure 2 to be self-contained, including the specific prompt/question associated with each task for better clarity.
>
> [1] Jansen, Petra, and Jennifer Lehmann. "Mental rotation performance in soccer players and gymnasts in an object-based mental rotation task." Advances in cognitive Psychology 9.2 (2013): 92.

---

> > ### Comment · Reviewer_PJzY · 2025-11-22
> >
> > 1. Question types like object relations are mainly studied in SAT [1] so I don't see why they are not genuine spatial reasoning.
> > 2. Are you using the exactly same dataset that was used for evaluating humans? If not, your paper should include human performance since MRT tasks are not trivial task for humans.

---

### Official Review · Reviewer_YfVR · 2025-11-01

**Soundness:** 3
**Presentation:** 3
**Contribution:** 2
**Rating:** 4
**Confidence:** 5

**Summary:**

This paper presents SRBench, a comprehensive diagnostic framework for evaluating spatial reasoning in Vision-Language Models (VLMs).
Building upon paradigms from cognitive psychology, the benchmark decomposes spatial reasoning into four pillars:
(1) Mental Rotation, (2) Spatial Visualization (Paper Folding), (3) Relational Understanding, and (4) Egocentric Navigation & Orientation.

The authors evaluate 17 state-of-the-art VLMs (including GPT-4o, QwenVL2.5, InternVL-3, LLaVA, Gemma, and MiniCPM) under synthetic and real-world conditions.
Findings show that while these models perform well on static spatial tasks (relations, orientation), their accuracy drops to near random on tasks requiring dynamic transformations such as rotation or folding. Scaling up model parameters improves transparency and reasoning articulation but does not overcome fundamental deficits in spatial simulation.

The paper concludes that bridging this “static-dynamic gap” requires new architectural inductive biases that explicitly support simulation-style spatial reasoning.

**Strengths:**

1. Systematic Framework：
The benchmark covers multiple cognitive dimensions, offering a holistic view of spatial reasoning rather than isolated tasks.

2. Insightful Findings:
The identified “static competence → dynamic collapse” phenomenon highlights a fundamental limitation in current VLMs and provides valuable diagnostic insight.

3. Clear, Engaging Presentation:
The writing is fluent and well-organized, with convincing examples and thoughtful analysis of scaling effects and failure modes.

**Weaknesses:**

**Lack of conceptual coherence across sub-tasks:**

The benchmark merges a diverse set of spatial reasoning tasks—mental rotation, paper folding, navigation, relational understanding—yet offers no clear theoretical or empirical justification for treating them as components of a single construct. Each of these capabilities has already been explored in specialized prior works, often with more controlled task design and deeper analysis. If the paper’s goal is diagnostic comprehensiveness, it should at least demonstrate how these tasks correlate or capture shared latent factors of spatial cognition, rather than serving as an unstructured “collection of puzzles.”

Without such analysis (e.g., inter-task correlation, factor analysis, or shared failure patterns), the paper reads as a broad but shallow aggregation—a “jack of all trades, master of none.” The integration would be more convincing if the authors could empirically show that performance across these tasks reflects a coherent spatial reasoning ability, rather than coincidental co-location of unrelated evaluations.


[1] *DOES SPATIAL COGNITION EMERGE IN FRONTIER MODELS?*
[2] *11Plus-Bench: Demystifying Multimodal LLM Spatial Reasoning with Cognitive-Inspired Analysis*
[3] *SpatialViz-Bench: An MLLM Benchmark for Spatial Visualization*
[4] *OmniSpatial: Towards Comprehensive Spatial Reasoning Benchmark for Vision Language Models*
[5] *Defining and Evaluating Visual Language Models' Basic Spatial Abilities: A Perspective from Psychometrics*

**Questions:**

On task integration:
You argue for a unified diagnostic framework of spatial reasoning, yet most cross-task analyses are missing. The only explicit discussion of inter-task relations appears in Section 3.3.2 (“The Fragility of Abstract Spatial Transformation”), which links Paper Folding and MRT tasks.
Could the authors clarify whether this relationship generalizes to the other sub-tasks (Relations, Orientation, Navigation)?
In other words, is there any empirical evidence that these tasks measure a shared latent spatial reasoning ability, or are they simply co-located evaluations?

---

> ### Author Response · Authors · 2025-11-20
>
> We thank the reviewer for the feedback. The systematic framework was designed to provide a holistic view of spatial reasoning, and the "static competence → dynamic collapse" phenomenon indeed represents a fundamental limitation in current VLMs that we believe offers valuable diagnostic insight. Below we address the weaknesses mentioned.
>
>
> **Weaknesses:**
>
> We respectfully disagree with the characterization of our benchmark as an "unstructured collection of puzzles." Our tasks were deliberately selected based on foundational cognitive psychology literature identifying mental rotation, visualization, relational reasoning, and navigation as core facets of human spatial intelligence. Nevertheless, we will add empirical demonstrations to further strengthen our position. We acknowledge that our differentiation from related work should be clearer. We will strengthen our positioning in the revision.
>
> **Questions:**
>
> The reviewer's core point—that we failed to empirically demonstrate the link between these tasks—is well-taken. Our initial submission made this connection on theoretical grounds but lacked the data to support it. To address this, we will perform the inter-task correlation analysis the reviewer suggests.

---

### Official Review · Reviewer_tUNJ · 2025-11-02

**Soundness:** 2
**Presentation:** 3
**Contribution:** 2
**Rating:** 4
**Confidence:** 4

**Summary:**

The paper analyzes the spatial reasoning capabilities of vlms, presenting a novel diagnostic benchmark to evaluate foundational aspects of spatial cognition. The authors focus on four core aspects: mental rotation, spatial visualization, relational understanding, and egocentric navigation to construct a systematic framework, which is based on cognitive psychology. Empirical results reveal that while VLMs can reliably handle static spatial relations and orientations, their performance drops to near-random levels on tasks requiring internal simulation or dynamic spatial manipulation (such as mental rotation and sequential paper folding), highlighting a significant gap in current VLM capabilities.

**Strengths:**

1. The paper introduces a diverse spatial reasoning benchmark, rigorously formulated from cognitive psychology. The benchmark systematically covers dynamic and static spatial reasoning, extending well beyond existing datasets that often focus on localized or single-facet assessments.
2. The paper identifies a key aspect of VLMs: they handle static spatial relations well but struggle with transformation-rich spatial reasoning, which is very insightful.

**Weaknesses:**

1. The paper convincingly identifies the performance gap, but does not provide a formal mathematical framework for why VLM architectures fail in dynamic spatial reasoning. For example, Section 3 attributes the collapse to a “lack of inductive biases” and hypothesizes about causes, but provides no detailed model analysis (e.g., explicit examination of attention map, model arch, visual embedding/encoding, how vision tokens represent transformations in latent space, or quantitative ablation of positional encoding mechanisms). While empirical, the absence of targeted analysis on the exact architectural or representational bottlenecks means the diagnosis stops short of design insight.
2. There are many spatial reasoning benchmarks for VLMs such as VSI-Bench, ViewSpatialBench, MindCube, and STARE and several settings (e.g., mental simulation and spatial understanding) are very similar to those in this paper. It would be helpful if the authors listed the differences from these benchmarks and discussed in detail how this work differs.
3. The paper evaluates only GPT-4o and o1 （proprietary model), which is not sufficient. To my knowledge, GPT-5 is released before the ICLR submission deadline and shows substantial gains in spatial understanding. The authors should include GPT-5 in the evaluation; otherwise, the results on older models may not reflect the capabilities of current frontier VLMs. For completeness and fairness, it would also be important to evaluate the latest Gemini and Claude VLMs.
4. One of the most valuable roles of a benchmark paper is to expose concrete failure modes and point to actionable paths for improvement. This paper concludes that “human-like reasoning will require not just greater scale, but new architectural paradigms and inductive biases” for dynamic object transformations. That insight is important, but it remains high-level. Could the authors substantiate it with targeted experiments that test specific, more concrete ideas for improving performance?

**Questions:**

1. Did the authors prepare different prompt templates and test VLM performance across prompt variants?
2. What is the human performance on each SRBench subtask?

---

> ### Author Response · Authors · 2025-11-20
>
> We thank the reviewer for acknowledging our work on the formulation of the benchmark and contribution on how VLMs handle static and dynamic tasks. Below we address the weaknesses mentioned.
>
>
> **Weaknesses:**
>
> 1. As a benchmark paper, our primary contribution is identifying and characterizing critical spatial reasoning gaps, providing the diagnostic tool that enables deeper architectural analyses. However, we agree that Section 3's empirical attributions (e.g., lacking inductive biases) would benefit from mechanistic evidence such as attention map analysis or ablations on positional encodings. We will add such probes and cite relevant work [1,2] that provides insights into these failure mechanisms.
> 2. Several works the reviewer mentions focus on different modalities (video rather than image), while others are  should be considered concurrent work. However our distinctions could be more immediately apparent, we will revise the manuscript accordingly
> 3.	We agree that evaluating state-of-the-art models is important and are currently updating our experiments to include the latest models from Google, OpenAI, and Anthropic. However, we note that proprietary models accessed via paywalls offer limited scientific value: even if they perform better, their closed nature prevents understanding why, limiting actionable insights for the community. Our focus remains on open models that enable reproducible research and meaningful improvements.
> 4.	This is a valuable suggestion. Our conclusion that paradigm shifts beyond scaling are needed (e.g., incorporating explicit spatiotemporal priors) was intentionally framed at a high level to highlight the benchmark's broader implications without overcommitting to untested specifics. We will add citations to work demonstrating current paradigm limitations and concrete directions for future research in the revised manuscript.
>
> **Questions:**
>
> 1.	We did not conduct a systematic prompt sensitivity analysis in our initial submission, using only a single template per task. We will run this analysis for our revision and report on performance variance, as this is a key factor in model robustness.
> 2.	We have not performed human evaluation on the specific benchmark. The benchmark was inspired from cognition tests performed on human to evaluate spatial understanding. Taking from that literature the average performance on these tests is around 70% which is significantly higher the average of the VLMs (30%)
>
> [1] Chen, Shiqi, et al. "Why is spatial reasoning hard for vlms? an attention mechanism perspective on focus areas." arXiv preprint arXiv:2503.01773 (2025).
> [2] Qi, Jianing, et al. "Beyond semantics: Rediscovering spatial awareness in vision-language models." arXiv preprint arXiv:2503.17349 (2025).

---

### Official Review · Reviewer_oEEL · 2025-11-07

**Soundness:** 2
**Presentation:** 2
**Contribution:** 1
**Rating:** 2
**Confidence:** 4

**Summary:**

The paper introduces SRBench, a diagnostic benchmark designed to assess dynamic spatial reasoning in VLMs. Building on paradigms from cognitive psychology (mental rotation, paper folding, spatial relations, egocentric orientation, navigation), the authors evaluate 17 SOTA models—including GPT-4o, InternVL-3, and Qwen-VL—across synthetic and naturalistic settings. Results show that while VLMs perform reasonably on static spatial tasks, they fail dramatically on dynamic transformations, often near random-chance accuracy. The paper argues that scaling and instruction tuning alone do not yield genuine spatial reasoning, calling for architectures with explicit inductive biases toward spatial simulation.

**Strengths:**

1. Timely and important problem. Spatial reasoning is a known blind spot in multimodal AI; the study provides systematic evidence of this deficiency.

2. Readable, well-presented results and qualitative analyses. Clear tables and insightful failure analyses enhance interpretability.

**Weaknesses:**

1. Although the suite spans four task families, each split is tiny (MRT-Easy=200, MRT-Hard=200, Paper-Folding=200, Relations=400, Orientation=400; Navigation size not specified in text). At ~1–2K total items, this is far below recent large-scale diagnostics and limits conclusions about generalization and fine-tuning. Report exact per-split counts (incl. Navigation), and either scale the corpus or narrow the paper’s claims.

2. The model set omits several state-of-the-art commercial VLMs (e.g., GPT o3/5, Google Gemini 2.5 pro), which weakens the headline claim about “current VLMs.”  They have more robust and powerful ability.

3. The paper provides excellent quantitative results but would benefit from a more in-depth qualitative error analysis. For instance, in the most difficult  task category, what are the specific failure modes? Do models fail at mental rotation differently than at geometric pattern recognition? Including a few examples of incorrect model outputs and analyzing the flawed reasoning (or lack thereof) would provide the community with a much richer understanding of the core challenges.

4. The scope is explicitly limited to four abilities (mental rotation, spatial visualization, relations, egocentric navigation). That misses key facets of spatial reasoning (e.g., 3D metric localization, occlusion/containment dynamics, multi-object compositional reasoning, frame-of-reference switching).

5. The paper fails to explicitly position itself relative to prior spatial reasoning benchmarks, such as SpatialVLM  SpatialRGPT and Omnispatial. These works already evaluate spatial understanding across text-only, multimodal, and cognitive-psychology-inspired paradigms.
However, the authors provide no systematic comparison table, no shared task taxonomy, and no empirical or conceptual justification for how their benchmark probes fundamentally new dimensions (e.g., integration, transformation, or dynamic reasoning) beyond what prior datasets test.

**Questions:**

I am curious about the true value of the dataset, as I strongly suspect that it may merely overfit to its own format.

If a base model (e.g., Qwen-VL) could be fine-tuned on this dataset and subsequently demonstrate performance gains on other benchmarks (such as VSI-Bench, OmniSpatial and SPACE), I would be much more inclined to recognize the dataset’s contribution.

---

> ### Author Response · Authors · 2025-11-20
>
> We thank the reviewers for their noted observations regarding the paper's presentation and insights on interpretability of model failures. Below we address the weaknesses mentioned.
>
> **Weaknesses:**
>
> 1.	We acknowledge that the Navigation task size (400 examples) was not clearly stated and will clarify this in the manuscript. At 1,8k examples total, our benchmark is smaller than some recent diagnostics. However, our scale aligns with established spatial reasoning benchmarks [1,2] that prioritize diagnostic precision. We will revise the paper to position our work explicitly as a targeted diagnostic tool for identifying specific spatial reasoning capabilities in VLMs and narrow our claims accordingly to reflect what this sample size supports.
> 2.	We acknowledge this limitation and will include recent models in the revision. However, we note that proprietary models accessed via paywalls offer limited scientific value: even if they perform better, their closed nature prevents the community from understanding why or building upon these insights. Our focus on analyzing open models enables reproducible research and actionable improvements for the broader community.
> 3.	Our initial submission focused on quantitative results as these models are Language-dominant by definition. However, we agree that a qualitative error analysis on specific failure modes is essential for understanding why models fail.
> 4.	We will add clarification that our benchmark targets low-level spatial visualization and transformation skills, which form the building blocks for more complex spatial reasoning tasks. Phenomena such as occlusion or object permanence rely on the same underlying operations (e.g., mental rotation, spatial relational reasoning). Thus, while the tasks differ in surface form, the cognitive substrate is shared, and our benchmark evaluates this substrate.
> 5.	We acknowledge that with the growing number of benchmarks, our distinctions could be more immediately apparent. We will add the suggested visualization to make our unique contributions clearer to readers.
>
> **Question:**
>
> It appears to be a misconception from the reviewer's perspective; we propose a new benchmark to evaluate VLM performance on the core facets of spatial reasoning and not a large-scale training corpus to finetune a model on. We do not claim that our evaluation suite provides information to teach model to reason about spatial structures and we also mention that we do not believe that new data is the solution.
>
>
> [1] Chen, Boyuan, et al. "Spatialvlm: Endowing vision-language models with spatial reasoning capabilities." Proceedings of the IEEE/CVF Conference on Computer Vision and Pattern Recognition. 2024.
> [2] Cheng, An-Chieh, et al. "Spatialrgpt: Grounded spatial reasoning in vision-language models." Advances in Neural Information Processing Systems 37 (2024): 135062-135093.

---

### Author Response · Authors · 2025-12-02
**Summary of Revisions and Response**

We thank the reviewers for their constructive feedback and the opportunity to revise our manuscript. We have significantly updated the paper to address concerns regarding the conceptual grounding, failure analysis, and comparative positioning of our work.
Below is a summary of the key changes made and a rationale for the experimental scope regarding points we believe are outside the immediate goals of this diagnostic study.

### 1. Major Revisions & Addressed Weaknesses

**Conceptual Coherence & Task Integration (Addressing Reviewer YfVR)** A primary critique was that the benchmark appeared to be a "collection of puzzles" without a unifying theoretical basis.
* **Action:** We have introduced **Section 3.1: Empirical Coherence of the Benchmark Tasks**. This includes a new hierarchical clustering analysis and correlation matrix (Figure 3) demonstrating that our sub-tasks (MRT, Paper Folding, Navigation, etc.) represent shared latent factors of spatial cognition rather than isolated tests. This empirically validates the psychometric grounding of the suite.

**Positioning Relative to SOTA Benchmarks (Addressing Reviewers oEEL, tUNj, & PJzY)** Reviewers requested better differentiation from concurrent works like *VSI-Bench* and *OmniSpatial*.
* **Action:** We have revised the **Related Work** section to more clearly delineate how SRBench uniquely targets the *Static-Dynamic Dissociation* in spatial reasoning, unlike other benchmarks that conflate perception and simulation or focus on video inputs.

**Clarification of Task Design (Addressing Reviewer PJzY)**
* **Action:** We have updated **Figure 2** and added a section **Appendix B** to provide clear, high-resolution examples of the prompts and visual inputs for every sub-task, resolving any ambiguity.

### 2. Rationale for Scope & Unaddressed Points

**Model Selection (Regarding Reviewers oEEL & tUNj)** Reviewers requested the inclusion of specific proprietary models (GPT-5, Gemini 2.5 Pro, GPT-o3).
* **Response:** While we acknowledge the rapid pace of model releases, we prioritised a comprehensive evaluation of **17 distinct models**, significantly expanding our open-source evaluation to include the latest **InternVL-3 (up to 78B)**, **Qwen2.5-VL**, and **Gemma 3** families. Additionally, whilst we initially intended to include the requested proprietary models (e.g., GPT-5)—and may have indicated this intention in our individual responses to reviewers—the prohibitive API costs associated with these new releases ultimately made their inclusion unfeasible for this study.
* **Defence:** We chose to focus heavily on these high-performing open-weights models to ensure the **reproducibility** of our diagnostic findings. While proprietary APIs (like GPT-5) offer high performance, their opacity and frequent updates make it difficult to pinpoint whether improvements stem from architectural reasoning capabilities or data contamination. Our results with GPT-4o and o1 already demonstrate that even proprietary frontiers struggle with the *Static-Dynamic* gap.

**Dataset Size (Regarding Reviewer oEEL)** Reviewer oEEL noted the dataset (~1,800 examples) is small compared to training benchmarks.
* **Response:** We have clarified the exact split counts in Section 2.
* **Defence:** We respectfully argue that SRBench is designed as a **diagnostic probe**, not a training corpus. In psychometrics, the validity of a test comes from the controlled isolation of variables, not the volume of samples. A sample size of ~1,800 is statistically sufficient to demonstrate the near-random performance of VLMs on mental rotation and folding tasks. Scaling the dataset would not change the fundamental finding that these models currently lack spatial reasoning.

**Theoretical Formalism & Internal Mechanisms (Regarding Reviewers oEEL & tUNj)** Reviewers suggested a need for formal mathematical frameworks or internal mechanistic analysis (e.g., attention maps) to explain the observed failures.

* **Response:** We respectfully argue that further analysis into internal architectural mechanisms is not in the scope of this study. The primary goal of SRBench is to diagnostically isolate the behavioural phenomenon—the Static-Dynamic Dissociation—which was previously obscured by confounding variables in other benchmarks. Investigating the internal causes (e.g., positional encoding failures) requires a distinct experimental framework. We have provided robust empirical examples of the failure modes to ground the phenomenon, and we reserve the causal architectural analysis for future work.

### Conclusion

We believe the revised manuscript now offers a robust, empirically grounded warning to the field: that scaling alone is not solving the problem of dynamic spatial simulation. We hope the Area Chair considers the value of this diagnostic insight, even if the scale of the dataset is intentionally kept focused to ensure psychometric rigour.

Sincerely,
The Authors

---

### Meta-Review · Area_Chair_9RY8 · 2026-01-01

**Summary:**

Reviewers raised two primary concerns: similarity to other benchmarks, and lack of coherence of the sub-tasks presented in this benchmark. The related work section and overall framing of the paper do not provide sufficient distinction from other popular benchmarks in the spatial reasoning domain, and the authors have not convincingly demonstrated during the rebuttal that their work is distinct. This is the primary reason for the rejection recommendation.

**Reviewer Concerns:**

### Addressed by the rebuttal
* Correlation analysis to show how sub-tasks are related.
* A slightly improved related work section incorporating a subset of the references provided by reviewers.

### Still outstanding:
* Fundamentally, a major issue lies with the similarity of this benchmark to others that already exist. The writing is not coherent around the static-dynamic point, and several aspects of the benchmark do not require this static-dynamic integration which the authors have pivoted the rebuttal around. As a result, this most important issue remains outstanding.

**Reviewer Scores:**

I do not believe any reviewer would have updated their score.

Reviewer oEEL
* no qualitative results were provided as requested.
* current VLMs were not tested as requested.
* no visualization contextualizing this benchmark with respect to others was provided. The new related work section does not explicitly differentiate the current work from prior works.

Reviewer tUNJ
* different prompt templates were not tested in the rebuttal, despite the importance being acknowledged by the authors.

Reviewer YfVR
* most concerned with sub-task coherence. Authors included hierarchical clustering analysis, but this analysis shows that the sub-tasks are *not* directly overlapping, and instead seem to capture different aspects of spatial understanding.

Reviewer PJzY
* figure 2 was not updated to be self-contained
* authors did not respond to follow-up questions from reviewer.

---

### Decision · Program_Chairs · 2026-01-26

Reject